# Cell-permeable succinate prodrugs rescue mitochondrial respiration in cellular models of acute acetaminophen overdose

Sarah Piel[1,2,3]*, Imen Chamkha[1,2], Adam Kozak Dehlin[1], Johannes K. Ehinger[1,2], Fredrik Sjövall[1,4], Eskil Elmér[1,2,5], Magnus J. Hansson[1,2]

1 Department of Clinical Sciences Lund, Mitochondrial Medicine, Lund University, Lund, Sweden, 2 NeuroVive Pharmaceutical AB, Medicon Village, Lund, Sweden, 3 Department of Anesthesiology and Critical Care Medicine, Center for Mitochondrial and Epigenomic Medicine, The Children's Hospital of Philadelphia, Philadelphia, United States of America, 4 Department of Clinical Sciences Lund, Skane University Hospital, Intensive Care and Perioperative Medicine, Lund University, Malmö, Sweden, 5 Department of Clinical Sciences Lund, Skane University Hospital, Clinical Neurophysiology, Lund University, Lund, Sweden

* sarah.piel@med.lu.se

**Data Availability Statement:** All relevant data are within the manuscript and its supplementary figure.

## Abstract

Acetaminophen is one of the most common over-the-counter pain medications used worldwide and is considered safe at therapeutic dose. However, intentional and unintentional overdose accounts for up to 70% of acute liver failure cases in the western world. Extensive research has demonstrated that the induction of oxidative stress and mitochondrial dysfunction are central to the development of acetaminophen-induced liver injury. Despite the insight gained on the mechanism of acetaminophen toxicity, there still is only one clinically approved pharmacological treatment option, N-acetylcysteine. N-acetylcysteine increases the cell's antioxidant defense and protects liver cells from further acetaminophen-induced oxidative damage. Because it primarily protects healthy liver cells rather than rescuing the already injured cells alternative treatment strategies that target the latter cell population are warranted. In this study, we investigated mitochondria as therapeutic target for the development of novel treatment strategies for acetaminophen-induced liver injury. Characterization of the mitochondrial toxicity due to acute acetaminophen overdose *in vitro* in human cells using detailed respirometric analysis revealed that complex I-linked (NADH-dependent) but not complex II-linked (succinate-dependent) mitochondrial respiration is inhibited by acetaminophen. Treatment with a novel cell-permeable succinate prodrug rescues acetaminophen-induced impaired mitochondrial respiration. This suggests cell-permeable succinate prodrugs as a potential alternative treatment strategy to counteract acetaminophen-induced liver injury.

## Introduction

Acetaminophen (paracetamol, N-acetyl-p-aminophenol; APAP) is one of the most common over-the-counter medications used worldwide [1, 2]. APAP is considered safe at therapeutic

**Funding:** This work was funded by Swedish government project and salary funding for clinically oriented medical research ALF-grant F 2014/354 to E.E., Regional research and development grants (Southern healthcare region, Sweden grant 170083 to E.E.;, The Crafoord Foundation grant 2017-0776 to E.E., and The Royal Physiographic Society in Lund grant 20141112 to M.J.H.). Additionally, this study was partially funded by NeuroVive Pharmaceutical AB (Lund, Sweden). NeuroVive Pharmaceutical provided support in the form of salaries for authors [S.P., J.K.E., I.C., E.E., and M.J. H.]. The specific roles of these authors are articulated in the 'author contributions' section. The funders did not have any additional role in the study design, data collection and analysis, decision to publish, or preparation of the manuscript.

**Competing interests:** S.P., J.K.E., I.C., F.S., E.E., and M.J.H have, or have had, salary from and/or equity interest in NeuroVive Pharmaceutical AB, a company active in the field of mitochondrial medicine. S.P., J.K.E., E.E., and M.J.H have filed patent applications for the use of succinate prodrugs for treatment of lactic acidosis or drug-induced side-effects due to complex I-related impairment of mitochondrial oxidative phosphorylation (WO/2015/155238) and protected carboxylic acid based metabolites for treatment of mitochondrial disorders (WO/2017/060400, WO/2017/060418, WO/2017/060422). This does not alter our adherence to PLOS ONE policies on sharing data and materials.

dose but has been associated with acute liver injury and liver failure in cases of intentional and unintentional overdose. In the western world, APAP accounts for up to 70% of acute liver failure cases [1–5]. Central to the development of APAP-induced liver injury is the formation of reactive oxygen species (ROS) and depletion of glutathione [6]. As a result, oxidative stress damages cellular proteins, including mitochondrial proteins, which induces further oxidative stress [1, 2, 6]. Within recent years, the critical role of mitochondrial function in the development of APAP-induced liver injury has been well established, but details on the exact mechanism of APAP's mitochondrial toxicity still remain controversial [2, 3, 6–8]. In addition, the majority of research was done in rodent models and the number of *ex vivo* or *in vivo* human studies addressing the mechanism of APAP-induced hepatotoxicity and the role of mitochondrial dysfunction are limited [9, 10]. Despite the extensive research that has been performed to date on APAP-induced liver failure, the only clinically approved pharmacological treatment option for APAP intoxication is N-acetylcysteine (NAC). NAC replenishes glutathione levels, increases the cell's antioxidant defense and thus, protects from further oxidative damage induced by APAP. It is more of preventive rather than rescuing nature, with lesser benefit for the already damaged cells [5, 7, 11]. Therefore, alternative treatment strategies that target the already damage liver cells are warranted.

In this this study, we investigated mitochondria as potential therapeutic target for treatment of APAP-induced liver injury *in vitro*. We first characterized the acute effect of APAP on mitochondrial function in primary human hepatocytes, HepG2 cells, and human platelets using respirometry. We then evaluated the efficacy of a cell-permeable succinate prodrug (NV241), a mitochondrially targeted alternative energy substrate, to rescue the impaired mitochondrial respiration following acute overdose of APAP.

## Materials and methods

### Materials

Unless otherwise stated, chemicals were purchased from Sigma-Aldrich Chemie GmbH (Schnelldorf, Germany). The cell-permeable succinate prodrug NV241 was provided by NeuroVive Pharmaceutical AB (Lund, Sweden) [12].

### Human liver cells

Human plateable primary hepatocytes (male, Caucasian, 69 years of age) were acquired from ThermoFisher Scientific (Bleiswijk, Netherlands) and plated as previously described [13].

The human hepatocyte carcinoma cell line HepG2 (male, Caucasian, 15 years of age) was purchased from Sigma-Aldrich Chemie GmbH (Schnelldorf, Germany). The cells were cultured at 37˚C and 5% $CO_2$ in minimum essential medium (MEM) (ThermoFisher Scientific, Bleiswijk, Netherlands) supplemented with 10% fetal bovine serum Sigma-Aldrich Chemie GmbH (Schnelldorf, Germany), 1% non-essential amino acids, 2 mM L-glutamine, 50 µg$^*$ml$^{-1}$ streptomycin and 50 U$^*$ml$^{-1}$ penicillin (all ThermoFisher Scientific, Bleiswijk, Netherlands). At 70–80% confluence, cells were collected using trypsin (0.05%, ThermoFisher Scientific, Bleiswijk, Netherlands), re-suspended in culture medium and counted using an automated cell counter (TC20™ Automated Cell Counter, Bio-Rad laboratories, Solna, Sweden) [13].

### Human platelets

The study was carried out in accordance with the Declaration of Helsinki. All blood cell experiments were performed with approval of the regional ethics committee of Lund University, Sweden (permit no. 2013/181). After written informed consent was acquired venous blood

from healthy volunteers was drawn in $K_2$EDTA tubes (Vacutainer®, BD, Franklin Lakes, USA) according to standard clinical practice. Human platelets were isolated and counted as previously described [14].

## Respirometry

Respiration of human primary hepatocytes was measured using the Seahorse XFe96 Analyzer (Agilent technologies, Massachusetts, USA). The day before the experiment, the primary hepatocytes were plated for four hours at 37˚C and 5% $CO_2$ at a cell density of 20 000 cells per well on collagen-coated 96-well plates (Agilent Seahorse XFe96 products, Agilent technologies, Waghaeusel-Wiesental, Germany). The plating medium was subsequently removed and replaced with culture medium of the same composition as for HepG2 cells. The cells were kept overnight at 37˚C and 5% $CO_2$ until use. Prior to the experiment the culture medium was replaced with XF-Base medium (Agilent Seahorse XF, Agilent technologies, Waghaeusel-Wiesental, Germany) containing 10 mM glucose, 2 mM L-glutamine and 5 mM sodium pyruvate (pH 7.4) and the cells were left to equilibrate for 1.5 hours at 37˚C and atmospheric $O_2$ and $CO_2$ until start of the respirometric protocol [13].

Mitochondrial respiration of the human carcinoma liver cell line HepG2 and of human platelets was measured with a high-resolution oxygraph (O2k, Oroboros Instruments, Innsbruck, Austria). Data were recorded using DatLab software versions 6 and 7 (Oroboros Instruments, Innsbruck, Austria) and respirometry was performed at 37˚C, with 2 mL active chamber volume and a stirrer speed of 750 rpm. Respirometry protocols with human platelets and HepG2 cells were performed in MiR05 medium (0.5 mM EGTA, 3 mM $MgCl_2$, 60 mM K-lactobionate, 20 mM Taurine, 10 mM $KH_2PO_4$, 20 mM HEPES, 110 mM sucrose and 1g/L bovine serum albumin) and all respiratory values were corrected for the oxygen solubility factor of the medium (0.92)[15]. Mitochondrial respiration was measured at cell concentrations of 200 x $10^6$ platelets per mL and 0.5 x $10^6$ HepG2 cells per mL [13, 14, 16, 17].

## Respirometric protocols for intact cells

The effect of APAP on mitochondrial respiration was first evaluated in intact human primary hepatocytes. Due to the restriction of 4 additions per sample in the Seahorse Analyzer increasing doses of APAP or vehicle were added to separate samples/wells. After routine respiration, the respiration dependent on oxidative phosphorylation of endogenous substrates, was measured, cells were exposed to vehicle (DMSO, control) or APAP (2.5, 5, 7.5 or 10 mM) for 15 min, followed by the addition of the protonophore carbonyl-cyanide p-(trifluoromethoxy) phenylhydrazone (FCCP, 1 μM) which was added to uncouple the electron transport system (ETS) from the phosphorylation pathway and measure maximal respiration dependent on the ETS alone. This was followed by a simultaneous addition of rotenone (2 μM) and the cell-permeable succinate prodrug NV241 (250 μM) to evaluate if mitochondrial complexes downstream of complex I (CI) are affected by APAP and if the cell-permeable succinate prodrug NV241 can bypass APAP-induced inhibition of mitochondrial respiration. Non-mitochondrial respiration was measured by addition of the complex III (CIII) inhibitor antimycin A (1 μg $^*$ $ml^{-1}$) and was subtracted from all respiratory values.

Next, the translatability of the human liver carcinoma cell line HepG2 and human platelets to study drug-induced mitochondrial and organ-specific toxicity was evaluated. HepG2 cells and human platelets were re-suspended in MiR05 and routine respiration was measured. After routine respiration stabilized, increasing, accumulative doses of APAP or vehicle (DMSO, control) were added to each sample. After the highest dose of APAP (10 mM) or vehicle was given, CI-linked mitochondrial respiration was inhibited by rotenone (2 μM) and the cell-

permeable succinate prodrug NV241 (250 μM) was added subsequently to investigate if mitochondrial complexes downstream of CI are affected by APAP and if the cell-permeable succinate prodrug NV241 can bypass APAP-induced inhibition of mitochondrial respiration. Non-mitochondrial respiration was measured by addition of antimycin A (1 μg * ml$^{-1}$), which all respiratory values were corrected for.

## Respirometric protocols for permeabilized cells

To further characterize the inhibitory effect of APAP on mitochondrial respiration, a substrate-uncoupler-inhibitor titration (SUIT) protocol was applied using HepG2 cells and human platelets. After routine respiration was measured, intact HepG2 cells and human platelets received either APAP (10 mM) or vehicle and were exposed for 10 min. Following the exposure, the plasma membrane was permeabilized using digitonin to allow substrates, which are otherwise impermeable, cellular access, followed by sequential additions of complex-specific substrates and inhibitors [17]. Platelets were permeabilized with 1 μg digitonin per 1$^*$10$^6$ platelets [14] and HepG2 cells were permeabilized with 10 μg digitonin per 1$^*$10$^6$ cells. The optimal digitonin concentrations were determined in separate experiments and found to induce maximal cell membrane permeabilization without disruption of mitochondrial respiration.

## Respirometric protocol to evaluate the coupling potential of the cell-permeable succinate prodrug NV241

In human platelets, the effect of APAP (10 mM) or vehicle (DMSO, control) on routine respiration was evaluated for 10 min, followed by the addition of the cell-permeable succinate prodrug NV241 (250 μM) or its vehicle (DMSO, control). Subsequently, coupled mitochondrial respiration, the respiration coupled to phosphorylation by the ATP-synthase, was measured by and calculated as the difference before and after addition of the ATP-synthase inhibitor oligomycin (1 μg/ml) [18]. The respirometric protocol was completed by measuring non-mitochondrial respiration following the addition of the CI inhibitor rotenone (2 μM) and the CIII-inhibitor antimycin A (1 μg/ml), which all respiratory values were corrected for.

## Data analysis

As the magnitude of change in the evaluated parameter was not pre-defined, power calculation for sample size was not applied. Experiments with HepG2 cells and human platelets were performed with a group size of six replicates and experiments with primary human hepatocytes were conducted with three separately prepared replicates of the same donor (each including ≤ 4 technical replicates per group). Statistical analyses were performed using Graph-Pad Prism version 7 (GraphPad Software, Inc., La Jolla, California, USA). Data are presented as mean ± range or scatter plot and mean ± range. Because the baseline routine respiration of primary hepatocytes demonstrated more variation before start of exposure to APAP as compared to HepG2 cells and human platelets, quantification and data analysis was performed with data expressed as percentage (%) of routine respiration (first measurement of routine respiration). All other data are expressed as pmol $O_2 \times sec^{-1} \times cell$ number$^{-1}$. Respiratory states measured by high-resolution respirometry of human platelets were previously found to be normally distributed [14], justifying the use of parametric tests for the present study. Analyses of differences between ≥3 groups were performed by one-way ANOVA with Dunnet's (Fig 1) or Tukey's (Fig 7) multiple comparison test. Paired, two tailed student's t-test was used for comparison of two groups (Figs 2, 3, 5 and 6). The half maximal inhibitory concentrations (IC$_{50}$)

were calculated by standard nonlinear curve fitting of normalized values (% of routine respiration, Fig 4). A p-value of 0.05 or less was considered to indicate significant differences.

## Results

### Effect of acetaminophen on mitochondrial respiration of intact primary hepatocytes, HepG2 cells and human platelets

We first assessed the effect of APAP on mitochondrial respiration in intact human primary hepatocytes. Following exposure to APAP for 15 min routine respiration was dose-

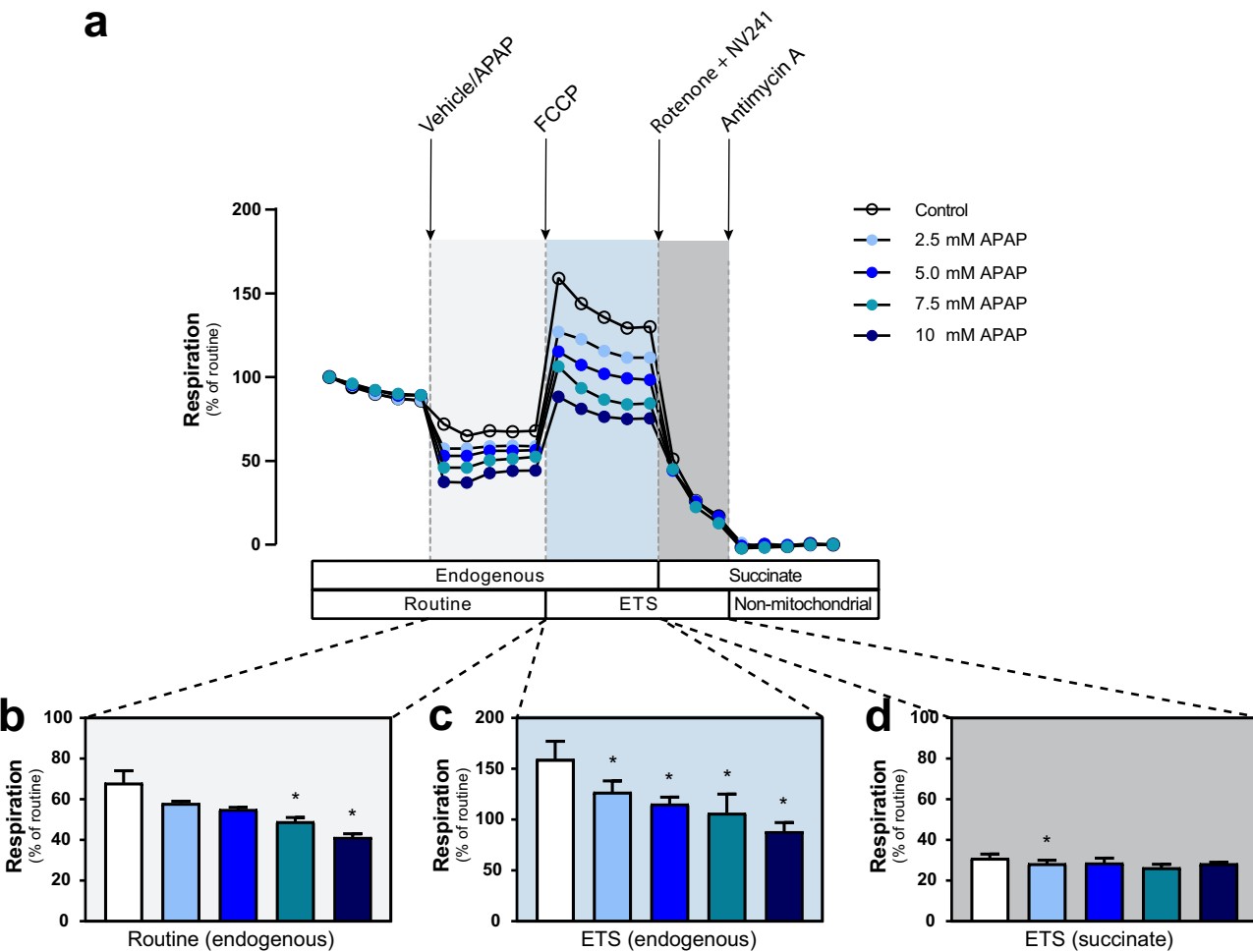

**Fig 1. Acetaminophen reduces mitochondrial respiration of human primary hepatocytes.** (a) Representative traces of simultaneously measured mitochondrial respiration of acetaminophen- (blue circles) or vehicle-treated (open circles) intact human primary hepatocytes. Increasing doses of acetaminophen (2.5, 5, 7.5 or 10 mM) were evaluated in separate samples and mitochondrial respiration was assessed by additions of the indicated specific mitochondrial uncouplers, inhibitors and substrates. The traces represent the mean of three independent experiments performed with hepatocytes from the same donor. Shaded background indicates the respiratory states shown in (b-d). (b-d) Quantification of the dose-response effect of acetaminophen (blue bars) or vehicle (control, open bar) on (b) routine respiration, (c) respiration uncoupled from oxidative phosphorylation pathways by FCCP and dependent on the electron transport system alone and (d) complex II-linked, succinate-dependent respiration assessed following subsequent simultaneous addition of the complex I inhibitor rotenone and the cell-permeable succinate prodrug NV241 to measure isolated complex II-linked mitochondrial respiration and evaluate the ability of NV241 to bypass acetaminophen-induced mitochondrial dysfunction. Data are expressed as mean plus range of the percentage (%) of routine respiration (b-d). One-way ANOVA with Dunnet's post-hoc analysis test was performed for analysis of differences. APAP: acetaminophen. ETS: electron transport system. FCCP: carbonyl-cyanide p-(trifluoromethoxy) phenylhydrazone. *$<$0.05. n = 3 from one donor.

dependently decreased compared to control (Figs 1 and 4). Subsequent uncoupling of the electron transport from the phosphorylation pathway with FCCP to measure mitochondrial respiration related to the ETS alone also showed a dose-dependent decrease with APAP as compared to vehicle control (Fig 1C). To evaluate whether mitochondrial complexes downstream of CI would be affected by APAP and if a cell-permeable succinate prodrug can bypass APAP-induced inhibition of mitochondrial respiration, a simultaneous addition of rotenone and the cell-permeable succinate prodrug NV241 followed. While the magnitude of decrease in mitochondrial respiration in response to this addition differed between vehicle-treated and APAP-intoxicated cells (Fig 1A), the respiration levels after the simultaneous addition of rotenone and NV241 were mostly similar between groups (Fig 1D). The remaining complex II (CII)-linked mitochondrial respiration supported by the cell-permeable succinate prodrug NV241 (Fig 1D) showed a minor difference between control and APAP-treated primary hepatocytes at the lowest dose tested (2.5 mM, $p < 0.05$) but no effect at higher doses, indicating a lack of dose-dependency.

Next, we assessed the suitability of the human hepatocyte carcinoma cell line HepG2 for in-depth characterization of the mitochondrial inhibition in liver cells induced by APAP. Similar to primary human hepatocytes, routine respiration supported by endogenous substrates decreased dose-dependently following exposure to APAP (Figs 2A and 4). At a dose of 10 mM, routine respiration was significantly reduced by 60% compared to vehicle control ($p < 0.01$) (Fig 2B). After addition of APAP, CI was inhibited with rotenone to subsequently measure CII-linked mitochondrial respiration in the presence of the cell-permeable succinate prodrug NV241 and isolated from any effects of APAP on CI and to additionally evaluate the ability of NV241 to bypass APAP-induced mitochondrial dysfunction. Addition of NV241 resulted in similar levels of CII-linked mitochondrial respiration in vehicle controls and APAP-intoxicated cells (Fig 2C).

We then evaluated the translatability of human platelets as surrogate tissue to study APAP's effect on mitochondrial function, using the same protocol as described for HepG2 cells. As primary, non-cultured human cells, human platelets from healthy donors present a source of viable, fresh mitochondria. In intact human platelets, routine respiration supported by endogenous substrates was likewise reduced dose-dependently in response to APAP (Figs 3A and 4). At the highest dose, APAP (10 mM) reduced routine respiration by 52% compared to vehicle control ($p < 0.001$) (Fig 3B). We continued the protocol with the addition of rotenone followed by the cell-permeable succinate prodrug NV241. Like in primary human hepatocytes and HepG2 cells, treatment with the cell-permeable succinate prodrug NV241 resulted in similar levels of respiration in vehicle controls and APAP-intoxicated cells (Fig 3C).

Despite differences in routine respiration before exposure, the sensitivity to inhibition by APAP was similar between the three cell types, with primary hepatocytes demonstrating a slightly lower $IC_{50}$ value than HepG2 cells and human platelets (primary hepatocytes: $IC_{50}$ 6.0 mM, HepG2 cells: $IC_{50}$: 6.6 mM and human platelets: $IC_{50}$: 7.4 mM, Fig 4). This demonstrates that HepG2 cells and human platelets are suitable cellular models for further evaluation of the inhibition of mitochondrial respiration by APAP.

## Characterization of the inhibition of mitochondrial respiration in HepG2 cells and human platelets

Further in-depth characterization of the inhibitory effect of APAP on mitochondrial respiration was performed using a Substrate-Uncoupler-Inhibitor-Titration (SUIT) protocol. After exposure to APAP (10 mM) for 10 min, intact platelets or HepG2 cells were permeabilized using digitonin, which was followed by sequential additions of complex-specific substrates and inhibitors at saturating concentrations to allow measurements of maximal respiratory

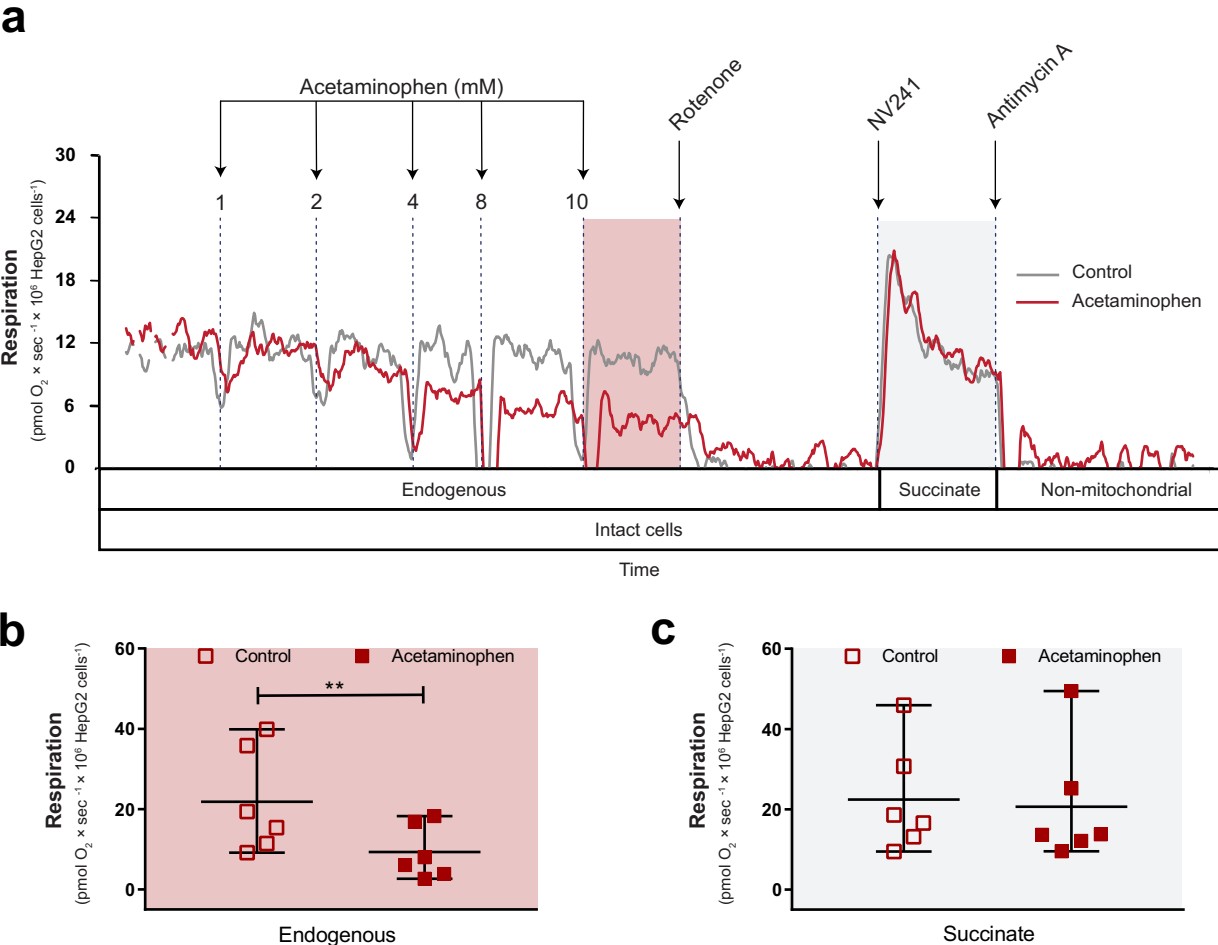

**Fig 2. Acetaminophen decreases mitochondrial respiration of intact HepG2 cells.** (a) Representative traces of simultaneously measured mitochondrial respiration of acetaminophen- (red trace) or vehicle-treated (control, grey trace) intact HepG2 cells. Increasing, accumulative doses of acetaminophen or vehicle (DMSO, control) were given, followed by inhibition of complex I with rotenone and subsequent addition of the cell-permeable succinate prodrug NV241 to assess complex II-linked, succinate-dependent respiration and evaluate the ability of NV241 to bypass acetaminophen-induced mitochondrial dysfunction. Non-mitochondrial respiration was measured by subsequent addition of the complex III inhibitor antimycin A, which all respiration values were corrected for. Boxes below the traces indicate the cellular state and substrate condition for each respiratory state and shaded background indicates the respiratory states shown in (b) and (c). Effect of vehicle (control, open square) or acetaminophen (10 mM, red square) on (b) routine respiration supported by endogenous substrates and (c) complex II-linked respiration in the presence of rotenone and the cell-permeable succinate prodrug NV241 in intact HepG2 cells. Data are expressed as individual scatter plots, mean and range. Paired, two tailed student's t-tests were performed for analysis of differences. ${}^{**}$p<0.01. n = 6.

capacities. Representative traces of simultaneously measured respiration of vehicle-treated and APAP-treated HepG2 cells are illustrated in Fig 5A.

In HepG2 cells, maximal CI-linked, ADP-stimulated mitochondrial respiration in the presence of the substrates malate, pyruvate, and glutamate (OXPHOS$_{CI-linked}$) was significantly decreased by 66% in APAP-treated cells (p<0.001) (Fig 5A and 5B). Despite decreased OXPHOS$_{CI-linked}$ respiration, convergent complex I+II (CI+II)-linked, maximal ADP-stimulated mitochondrial respiration in the presence of malate, pyruvate, glutamate, and succinate (OXPHOS$_{CI+II-linked}$) was unchanged in APAP-intoxicated HepG2 cells as compared to control (Fig 5A and 5C). Both maximal convergent CI+II- and CII-linked mitochondrial respiration dependent on the ETS alone were unaffected in HepG2 cells by APAP (S1 Fig).

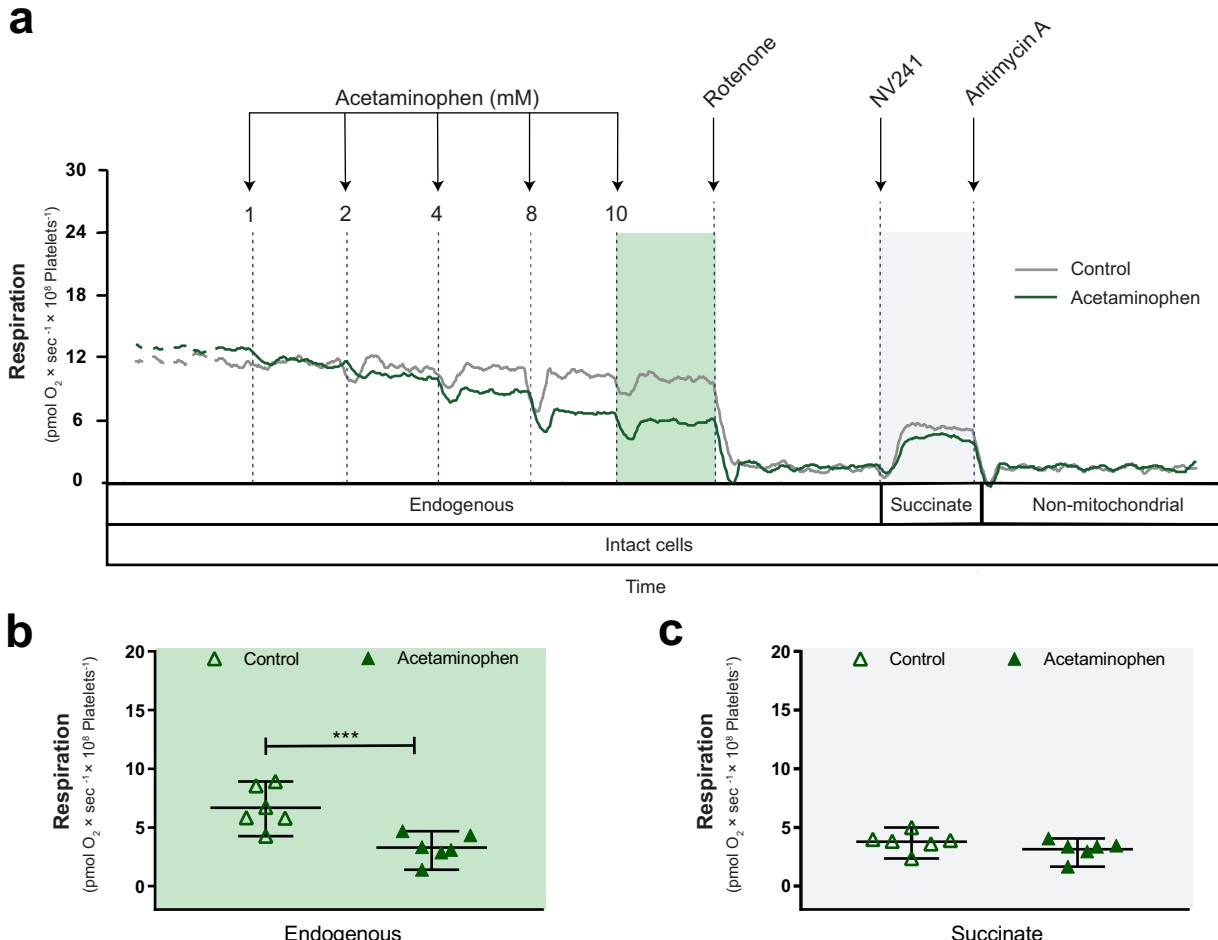

**Fig 3. Acetaminophen decreases mitochondrial respiration of intact human platelets.** (a) Representative traces of simultaneously measured mitochondrial respiration of acetaminophen- (green trace) or vehicle-treated (control, grey trace) intact human platelets. Increasing, accumulative doses of acetaminophen or vehicle (DMSO) were given, followed by inhibition of complex I with rotenone and addition of the cell-permeable succinate prodrug NV241 to assess complex II-linked, succinate-dependent respiration and to evaluate the ability of NV241 to bypass acetaminophen-induced mitochondrial dysfunction. Non-mitochondrial respiration was measured by subsequent addition of the complex III inhibitor antimycin A, which all respiration values were corrected for. Boxes below the traces indicate the cellular state and substrate condition for each respiratory state and shaded background indicates the respiratory states shown in (b) and (c). Effect of vehicle (control, open triangle) or acetaminophen (10 mM, green triangle) on (b) routine respiration supported by endogenous substrates and (c) complex II-linked respiration in the presence of rotenone and the cell-permeable succinate prodrug NV241 in intact human platelets. Data are expressed as individual scatter plots, mean and range. Paired, two tailed student's t-tests were performed for analysis of differences. ***$p < 0.001$. n = 6.

In human platelets, maximal CI-linked and convergent CI+II-linked, ADP-stimulated mitochondrial respiration, as well as maximal convergent CI+II-linked respiration dependent on the ETS alone was reduced following exposure to APAP: OXPHOS$_{\text{CI-linked}}$ ($p < 0.01$, Fig 6A), OXPHOS$_{\text{CI+II-linked}}$ ($p < 0.01$, Fig 6B) and ETS$_{\text{CI+II-linked}}$ ($p < 0.01$, Fig 6C), respectively. Like in HepG2 cells, maximal CII-linked respiration dependent on the ETS alone (ETS$_{\text{CII-linked}}$) remained unaffected by APAP (Fig 6D).

### Treatment effect of a cell-permeable succinate prodrug on acetaminophen-induced inhibition of mitochondrial respiration

Lastly, we evaluated if the normalization of mitochondrial respiration by this novel pharmacological treatment strategy is linked to phosphorylation activity by the ATP-synthase. This was

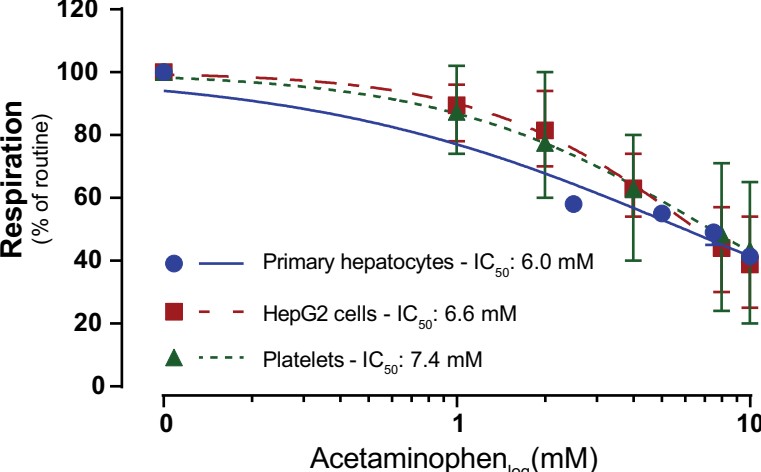

**Fig 4. Dose-response comparison of the inhibitory effect of acetaminophen on mitochondrial respiration of intact human primary hepatocytes, HepG2 cells and human platelets.** The dose-response effect of acetaminophen on routine respiration supported by endogenous substrates was determined in intact primary human hepatocytes, HepG2 cells and human platelets. Data are expressed as percentage (%) of routine respiration before addition of acetaminophen and are presented as mean plus range. Standard nonlinear curve fitting of the normalized values was applied to obtain half maximal inhibitory concentration ($IC_{50}$) values for the effect of acetaminophen on routine respiration in human primary hepatocytes (blue circle), HepG2 cells (red square) and human platelets (green triangle). Data are expressed as mean plus range. Primary hepatocytes: n = 3, from one donor. HepG2 cells: n = 6. Platelets: n = 6.

evaluated in intact human platelets following exposure to APAP (10 mM) for 10 min, with and without subsequent treatment, and calculated as the difference in respiration before and after the inhibition of the ATP-synthase. Mitochondrial respiration coupled to phosphorylation by the ATP-synthase, here referred to as coupled respiration, was decreased by 40% (p<0.01) by APAP (Fig 7). Treatment with the cell-permeable succinate prodrug NV241 rescued coupled respiration and restored it to the level of controls (Fig 7).

## Discussion

In this study, we demonstrated that APAP induces an immediate inhibition of mitochondrial respiration in human-derived cells through interference with CI or upstream metabolism while respiration associated with CII and downstream complexes remains unaffected. The toxicity profile of APAP on mitochondrial respiration was not exclusive to hepatic cells and was confirmed in fresh human platelets, presenting them as suitable surrogate tissue to study the role of mitochondrial dysfunction in acute APAP-induced toxicity. Treatment with a cell-permeable succinate prodrug normalized the drug-induced impairment of mitochondrial respiration, demonstrating the ability of succinate to bypass APAP-induced mitochondrial dysfunction and presenting cell-permeable succinate as a potential novel pharmacological treatment strategy for APAP-induced liver injury.

APAP is the main cause for acute liver failure in the western world and, with a mortality rate of 0.4%, not uncommonly ends fatally [1–4]. The critical role of mitochondrial dysfunction in the development of APAP-induced liver injury and failure has been previously reported by others [2, 3, 6, 7, 9, 10, 19]. Inhibition of the respiratory chain, induction of mitochondrial permeability transition, increased mitochondrial oxidative stress, decreased mitochondrial ATP production and increased mitophagy has been associated with APAP overdose [2, 3, 6, 7]. In this study, we demonstrated that APAP induces mitochondrial toxicity through or

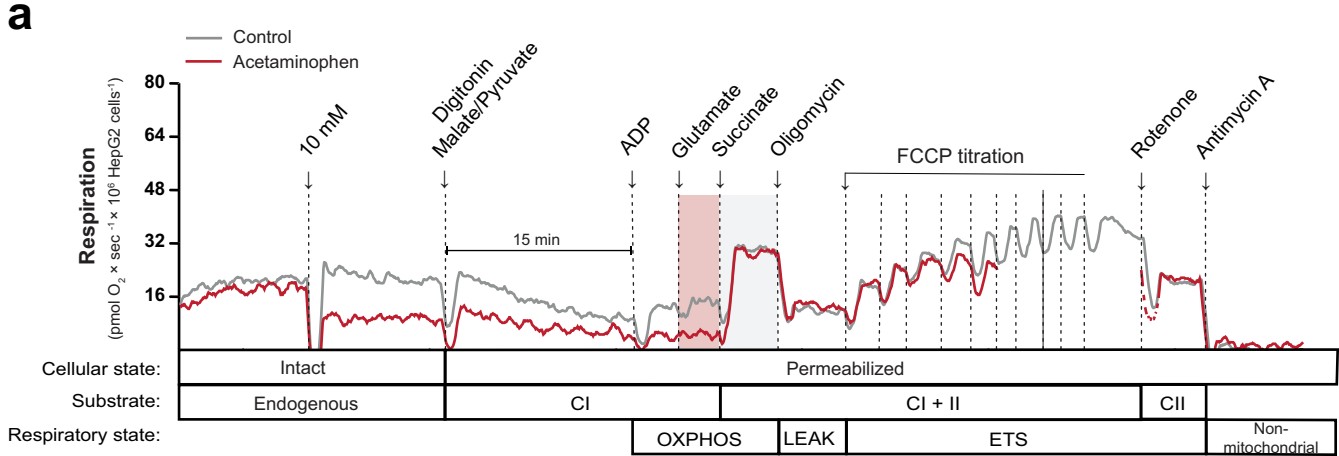

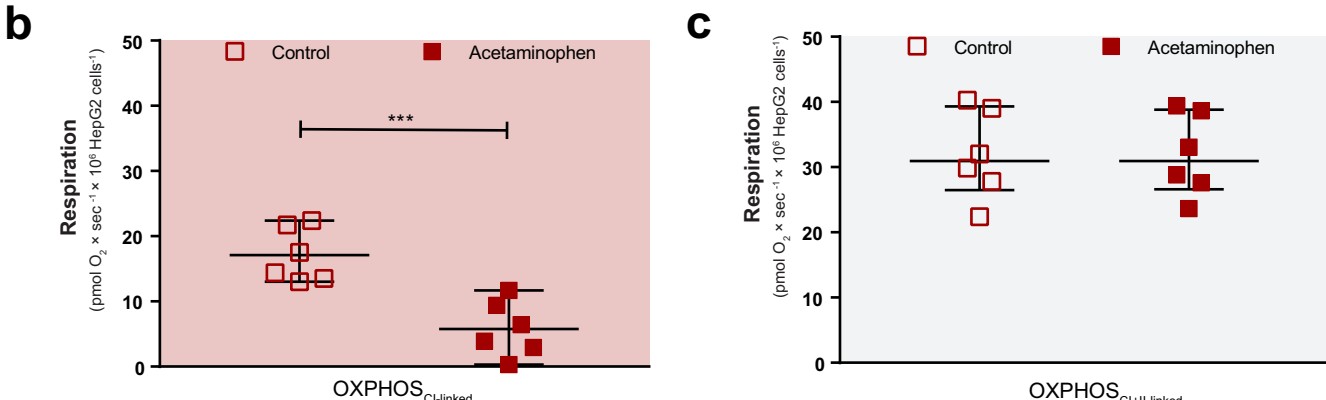

**Fig 5. Acetaminophen inhibits mitochondrial respiration of HepG2 cells through complex I.** (a) Representative traces of the Substrate-Uncoupler-Inhibitor-Titration (SUIT) protocol applied to acetaminophen- (10 mM, red trace) or vehicle-treated (control, grey trace) and subsequently permeabilized HepG2 cells. Maximal convergent CI+II-linked mitochondrial respiration, which was uncoupled from the phosphorylation pathway using FCCP and dependent on the electron transport system alone, was reached with fewer additions of FCCP in the acetaminophen-intoxicated sample. For comparison of respiration levels of both treatment groups, the trace overlay was adjusted based on the respiratory protocol instead of time. The immediate decrease in mitochondrial respiration following each addition is a temporary artefact due to the injection of oxygen with each compound solution, which creates a brief disturbance of the respiration signal, and which is unrelated to the compound added. Boxes below the traces indicate the cellular state, the respiratory complexes utilized for respiration during oxidation of the given substrates, as well as the respiratory states at the indicated parts of the protocol. Shaded background indicates the respiratory states shown in (b) and (c). (b) Maximal complex I-linked, ADP stimulated mitochondrial respiration ($OXPHOS_{CI-linked}$) and (c) maximal convergent complex I and II-linked, ADP stimulated mitochondrial respiration ($OXPHOS_{CI+II}$) of acetaminophen- (10 mM, red square) or vehicle-treated (control, open square) HepG2 cells. (b-c) Data are expressed as individual scatter plots and mean plus range. CI: complex I. CII: complex II. CI+II: complex I+II. ETS: electron transport system. FCCP: carbonyl-cyanide p-(trifluoromethoxy) phenylhydrazone. OXPHOS: oxidative phosphorylation. ***$p < 0.001$. n = 6.

upstream of CI. The results of this study are corroborated by a study recently published by Chrøis, Larsen, Pedersen, Rygg, Boilsen, Bendtsen et al. [20] demonstrating inhibition of CI-linked mitochondrial respiration by APAP using *ex vivo* human liver. By inhibiting CI-linked pathways, the most efficient way to oxidize NADH, translocate protons across the inner mitochondrial membrane, uphold the mitochondrial membrane potential and produce ATP is disabled by APAP [21]. The experimental design of our study and the study by Chrøis, Larsen, Pedersen, Rygg, Boilsen, Bendtsen et al. [20] included exposure of intact cells and tissues to APAP. This allows for the mitochondrial toxic effect to be caused by either APAP directly or N-acetyl-p-benzoquinone imine, that is the reported highly toxic metabolite of APAP which is

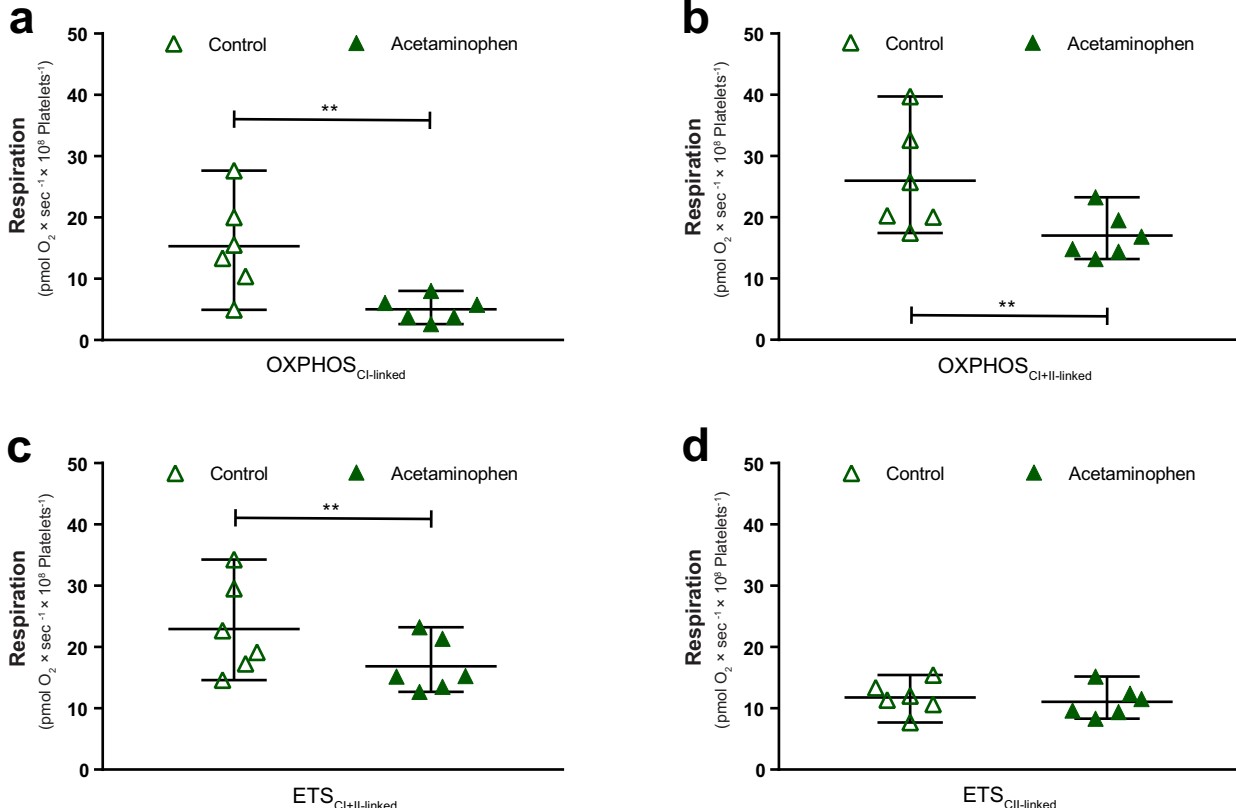

**Fig 6. Acetaminophen decreases mitochondrial respiration of human platelets through complex I.** Effect of the exposure of intact platelets to acetaminophen (green triangle) or vehicle (control, open triangle) and subsequent permeabilization of the plasma membrane to apply a Substrate-Uncoupler-Inhibitor-Titration protocol and assess (a) maximal complex I-linked, ADP stimulated mitochondrial respiration ($OXPHOS_{CI-linked}$)) and (b) maximal convergent complex I and II-linked, ADP stimulated mitochondrial respiration ($OXPHOS_{CI+II-linked}$), as well as (c) maximal convergent complex I and II-linked mitochondrial respiration dependent on the electron transport system alone ($ETS_{CI+II-linked}$) and (d) maximal complex II-linked mitochondrial respiration dependent on the electron transport system alone ($ETS_{CII-linked}$). Data are expressed as individual scatter plots and mean plus range. CI: complex I. CII: complex II. CI+II: complex I+II. ETS: electron transport system. OXPHOS: oxidative phosphorylation. $^{**}p<0.01$. n = 6.

generated intracellularly at excessive amounts when the APAP-induced oxidative stress has depleted cellular glutathione. Independent of the origin of the toxic species, CII or down-stream complexes were left mostly unaffected. The effect on CII-linked mitochondrial respiration observed in primary hepatocytes did not follow a dose-response pattern as only the lowest concentration of APAP tested showed a minor reduction of respiration. Therefore, the observed reduced CII-linked mitochondrial respiration in primary hepatocytes is likely unspecific and not related to APAP.

Currently, the only clinically approved pharmacological treatment option for APAP over-dose is NAC. NAC replenishes glutathione levels which increases the cell's ability to scavenge ROS. Thus, it protects liver cells from further APAP-induced oxidative injury [1, 5, 22]. Already damaged liver cells, however, benefit little from NAC treatment. Therefore, alternative treatment strategies are needed that can rescue the already damaged liver cells and prevent the resulting acute liver failure. At the preclinical stage, a limited number of mitochondrial tar-geted treatment strategies have shown success. The most promising pharmacological strategy, a mitochondrial-targeted antioxidant, decreased the magnitude of liver injury in mouse mod-els of late-stage presenting APAP intoxication by reducing mitochondrial-related ROS

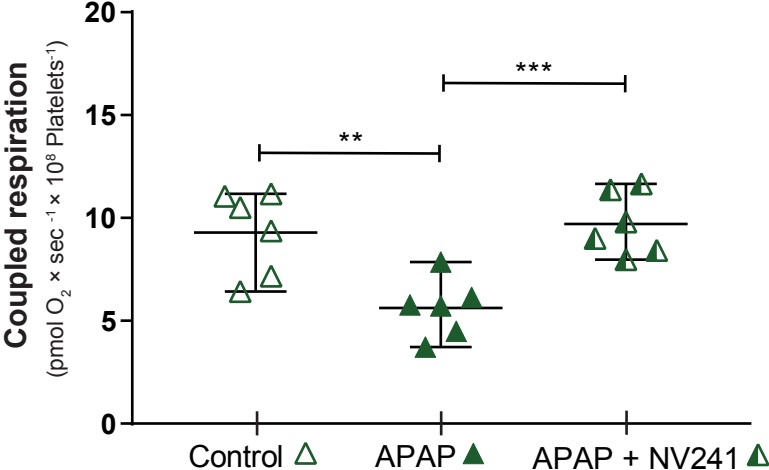

**Fig 7. The cell-permeable succinate prodrug NV241 rescues coupled respiration of intact human platelets following acute intoxication with acetaminophen.** Intact human platelets were exposed to acetaminophen (10 mM, green triangle) or vehicle (control, open triangle) for 10 min and subsequently treated with the cell-permeable succinate prodrug NV241 (half-filled green triangle). Coupled respiration, defined as the mitochondrial respiration linked to phosphorylation by the ATP synthase, was calculated as the difference in respiration before and after addition of the ATP-synthase inhibitor oligomycin. Data are expressed as mean plus range. One-way ANOVA with Tukey's post hoc test was performed for analysis of differences. APAP: acetaminophen. $^{**}p<0.01$. $^{***} = p<0.001$. n = 6.

production [7, 22, 23]. In this study, we demonstrated that CII-linked mitochondrial metabolism of a cell-permeable succinate prodrug can bypass and compensate for the decreased CI-linked mitochondrial metabolism following acute APAP exposure. These findings point towards a novel alternative treatment strategy for APAP-induced liver failure: mitochondrial-targeted, cell-permeable succinate prodrugs. Supplementation of an alternative energy source that liver cells can utilize despite the inhibitory effect of APAP on CI-linked metabolism could potentially allow them to maintain the required level of energy production and thus, rescue already injured liver cells. Succinate treatment has previously demonstrated by others to improve bioenergetics and reduce cell death *in vitro* in models of traumatic brain injury, metformin-induced and oxidant-induced mitochondrial dysfunction [24–26], thus, further supporting this hypothesis. The cell-permeable succinate prodrug presented in this study is the lead candidate of the first generation of an extensive rational drug design program focused around Krebs cycle intermediates for treatment of mitochondrial dysfunction and related disorders. The succinate prodrug has improved cell-membrane permeability over succinate and has shown to release succinate intracellularly, bypass mitochondrial complex I-related dysfunction and support oxidative phosphorylation [12, 18, 27]. Because NV241 lacks sufficient stability in plasma and serum containing media, we were not able to investigate its treatment effect on long-term cellular effects caused by APAP or *in vivo*. Currently, compounds which are more stable and suitable for *in vivo* use are under development for future studies [28].

Succinate has primarily been known as a metabolite of the TCA cycle. Over time, it has emerged to play a role in epigenetics, cell proliferation, paracrine signaling, ROS formation through reversed electron transport (RET) and inflammation [29–31]. While the risk of increased ROS through RET is low in the presence of CI inhibition [32], the role of succinate in inflammation, especially during APAP-induced liver damage, needs to be further investigated. In the liver, succinate has been shown to contribute to activation of hepatic stellate cells and Kupffer cells, which phagocytose dead and apoptotic parenchymal cells but also send out pro-inflammatory signals and thus, potentially further aggravate APAP-induced liver injury

[30, 33]. Whether a succinate-induced pro-inflammatory response would aggravate APAP-induced liver injury or stimulate tissue repair pathways remains to be elucidated [30, 34].

In this study, human platelets and the human carcinoma liver cell line HepG2 were used as surrogate tissues to study the effect of APAP on mitochondrial respiration as well as the rescue effect of NV241 in acute APAP intoxication. The translatability between human platelets and tissue specific cell lines is continuously reevaluated. Human platelets, a fresh source of viable mitochondria, have been described to rely on oxidative phosphorylation and reflect mitochondrial function of other, more metabolically active tissues [35–38]. Also cancer cells, long believed to rely solely on glycolysis, have now been described to upregulate their mitochondrial metabolism under certain conditions and rely on mitochondrial function for several cancerogenic processes [39, 40]. Even though there are important differences between primary hepatocytes, HepG2 cells and human platelets, our data indicate that they show a relative comparable sensitivity towards drug-induced mitochondrial dysfunction, as indicated by the similar $IC_{50}$ values determined in this study. Our data therefore present these cell types as suitable surrogate tissues to study the role of mitochondrial dysfunction in drug-induced toxicity and further indicate that the liver specific toxicity in patients with acute APAP intoxication is likely due to the first-pass metabolism of APAP instead of liver specific metabolism of the drug.

In conclusion, in this study we demonstrated, using human-derived cells, that APAP induces mitochondrial inhibition through CI (or upstream thereof) while CII and downstream complexes are unaffected. We further showed that a cell-permeable succinate prodrug normalizes APAP-induced inhibition of mitochondrial respiration, presenting pharmacological bypass of APAP-induced mitochondrial toxicity with cell-permeable succinate prodrugs as a promising alternative treatment strategy for APAP-induced mitochondrial dysfunction and, potentially, liver injury.

## Supporting information

**S1 Fig. Effect of acetaminophen on the electron transport system of HepG2 cells.** Effect of the exposure of intact HepG2 cells to acetaminophen (red square) or vehicle (control, open square) in subsequently permeabilized cells to apply a Substrate-Uncoupler-Inhibitor-Titration protocol and assess the effect of acetaminophen on mitochondrial respiration which was uncoupled from the phosphorylation pathway using FCCP and dependent on the electron transport system alone. (a) Maximal convergent complex I and II-linked mitochondrial respiration dependent on the electron transport system alone ($ETS_{CI+II-linked}$) and (b) maximal complex II-linked mitochondrial respiration dependent on the electron transport system alone ($ETS_{CII-linked}$). Data are expressed as individual scatter plots and mean plus range. CII: complex II. CI+II: complex I+II. ETS: electron transport system. FCCP: carbonyl-cyanide p-(trifluoromethoxy) phenylhydrazone. n = 6.
(EPS)

## Acknowledgments

The authors would like to thank Jonathan P. Starr for proofreading the manuscript.

## Author Contributions

**Conceptualization:** Sarah Piel, Johannes K. Ehinger, Fredrik Sjövall, Eskil Elmér, Magnus J. Hansson.

**Data curation:** Sarah Piel, Imen Chamkha, Adam Kozak Dehlin.

**Formal analysis:** Sarah Piel.

**Funding acquisition:** Eskil Elmér, Magnus J. Hansson.

**Investigation:** Magnus J. Hansson.

**Methodology:** Sarah Piel.

**Project administration:** Eskil Elmér, Magnus J. Hansson.

**Supervision:** Fredrik Sjövall, Eskil Elmér, Magnus J. Hansson.

**Writing – original draft:** Sarah Piel.

**Writing – review & editing:** Imen Chamkha, Adam Kozak Dehlin, Johannes K. Ehinger, Fredrik Sjövall, Eskil Elmér, Magnus J. Hansson.

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
