## [Decision Letter · Decision Letter 0]

28 Jan 2020

PONE-D-20-01070

Cell-permeable succinate prodrugs rescue mitochondrial respiration in cellular models of acute acetaminophen overdose

PLOS ONE

Dear Dr. Piel:

Thank you for submitting your manuscript to PLOS ONE. After careful consideration, we feel that it has merit but does not fully meet PLOS ONE’s publication criteria as it currently stands. Therefore, we invite you to submit a revised version of the manuscript that addresses the points raised during the review process.

We would appreciate receiving your revised manuscript by Mar 13 2020 11:59PM. To enhance the reproducibility of your results, we recommend that if applicable you deposit your laboratory protocols in protocols.io, where a protocol can be assigned its own identifier (DOI) such that it can be cited independently in the future. For instructions see: http://journals.plos.org/plosone/s/submission-guidelines#loc-laboratory-protocols

We look forward to receiving your revised manuscript.

Kind regards,

Jianhua Zhang

Academic Editor

PLOS ONE

Journal Requirements:

2. Please note that all PLOS journals ask authors to adhere to our policies for sharing of data and materials: https://journals.plos.org/plosone/s/data-availability. According to PLOS ONE’s Data Availability policy, we require that the minimal dataset underlying results reported in the submission must be made immediately and freely available at the time of publication. As such, please remove any instances of 'unpublished data' or 'data not shown' in your manuscript and replace these with either the relevant data (in the form of additional figures, tables or descriptive text, as appropriate), a citation to where the data can be found, or remove altogether any statements supported by data not presented in the manuscript.

'E.E.: Swedish government project and salary funding for clinically oriented medical research (ALF-grants; F 2014/354) URL: https://www.skane.se/en/politics-and-organisation/research/services-and-support-for-researchers/grants-and-applications/

E.E.: Regional research and development grants (Southern healthcare region, Sweden; 170083): URL: https://www.skane.se/en/politics-and-organisation/research/services-and-support-for-researchers/grants-and-applications/

E.E.:The Crafoord Foundation (2017-0776) URL: https://en.crafoord.se/

M.J.H.:The Royal Physiographic Society in Lund (20141112) URL: https://www.fysiografen.se/en/

The funders had no role in study design, data collection and analysis, decision to publish, or preparation of the manuscript.'

We note that one or more of the authors are employed by a commercial company: NeuroVive Pharmaceutical AB

We also note that you have a patent relating to material pertinent to this article.

In the amended statement of Competing Interests please declare this patent (with details including name and number), along with any other relevant declarations relating to employment, consultancy, patents, products in development or modified products etc.

Please confirm that this does not alter your adherence to all PLOS ONE policies on sharing data and materials, as detailed online in our guide for authors http://journals.plos.org/plosone/s/competing-interests by including the following statement: "This does not alter our adherence to  PLOS ONE policies on sharing data and materials.” If there are restrictions on sharing of data and/or materials, please state these. Please note that we cannot proceed with consideration of your article until this information has been declared.

Reviewers' comments:

Reviewer's Responses to Questions

**Comments to the Author**

1. Is the manuscript technically sound, and do the data support the conclusions?

Reviewer #1: Yes

Reviewer #2: Partly

2. Has the statistical analysis been performed appropriately and rigorously? 

Reviewer #1: Yes

Reviewer #2: Yes

3. Have the authors made all data underlying the findings in their manuscript fully available?

Reviewer #1: Yes

Reviewer #2: Yes

4. Is the manuscript presented in an intelligible fashion and written in standard English?

Reviewer #1: Yes

Reviewer #2: No

5. Review Comments to the Author

Reviewer #1: The manuscript entitled “Cell-permeable succinate prodrugs rescue mitochondrial respiration in cellular models of acute acetaminophen overdose” by Piel et al investigates the effects of acute acetaminophen exposure on mitochondrial function in primary human hepatocytes, HepG2 hepatocellular carcinoma cells, and platelets. The authors demonstrate that acetaminophen inhibits complex I, but not complex II function, and that treatment with a succinate prodrug (NV241) can rescue acetaminophen-induced mitochondrial dysfunction. Overall, the paper is very well written and clearly displays the negative respiratory outcomes associated with exposure to high concentrations of acetaminophen. The concept of utilizing succinate-based prodrugs to rescue pathologies associated with mitochondrial complex I dysfunction, particularly in an acute toxic exposure scenario, is an interesting area of research. There are a few minor points that should be addressed prior to publication.

1) One potential limitation in interpretation of the respiration graphs is that data are normalized to what I presume to be an initial seeding/plating density (i.e. cell number prior to treatment). Did the authors ever measure total protein or do some assessment of viability following acetaminophen and/or the mitochondrial inhibitor exposure? It would seem that the toxicity, particularly in the case of acetaminophen, could result in significant cell death from the starting number and thus influence the overall respiratory rates.

2) In Figure 1, why were rotenone and NV241 added at the same time, whereas in Figures 2 and 3 NV241 was always added post-rotenone exposure? It would seem that the minimal effect of NV241 in the primary hepatocytes may be a result of the cotreatment blocking the rescue effect.

3) It would seem that the reliance on oxidative phosphorylation might differ significantly between a primary hepatocyte, a liver cancer cell, and a circulating platelet. Did the authors take into account ECAR/glycolysis measurements in this or previous publications to account for changes in glycolytic flux? This could be covered in the discussion.

4) In Figure 2B, it appears that two of the chosen wells are accounting for the significantly higher respiration in control HepG2 cells compared to the acetaminophen treated cells (even with the indicated significant). The authors should consider removing outliers or adding in additional wells to ensure the significant decrease in respiration that occurs following acetaminophen exposure in these cells.

5) In Figure 5A, why does the FCCP titration taper off in the acetaminophen treated cells and then recover to control following rotenone treatment? Was this a result of negative readings/values? The authors could discuss in the results or discussion section.

Reviewer #2: While the manuscript submitted uses a variety of respirometry methods to accurately assess mitochondrial respiration (and therefore metabolism), as written, the manuscript makes conclusions that are "reaching" and not entirely supported by the data. Below are the major concerns that are outlined - these revision experiments would make the manuscript suitable for publication.

Major Concerns:

The authors present their succinate prodrug as an adjunctive therapy to NAC, yet do not do a single experiment comparing their prodrug to NAC, nor any combination therapy experiments. If the conclusion that this prodrug is adjunctive, they must show that in their experimental models.

The authors did not assess the effect of their prodrug on any endpoints apart from mitochondrial respiration. While this reviewer appreciates the rationale (that this prodrug is specifically rescuing mitochondrial respiration), it would be interesting to see how this drug impacts downstream endpoints of cellular health that are affected by acetaminophen toxicity, such as ROS production, apoptosis/necrosis, cell proliferation, etc.

Pursuant to the previous comments, assessing downstream cell health endpoints after combination therapy would greatly strengthen the authors conclusions.

Minor Concerns:

The manuscript is generally well-written, however it would benefit from a thorough edit for syntax, use of commas, plurals, etc.

6. PLOS authors have the option to publish the peer review history of their article (what does this mean?). If published, this will include your full peer review and any attached files.

Reviewer #1: No

Reviewer #2: No

---

## [Author Response · Author response to Decision Letter 0]

8 Mar 2020

Dear editor,

thank you for inviting us to resubmit our manuscript (PONE-D-20-01070). We are grateful for the constructive comments from the editor and the reviewers. We have carefully addressed all their concerns. The editors and reviewers’ comments are displayed below in normal font style and the specific response to each issue raised in the comments is displayed italic font style.

For the authors,

Sarah Piel

Journal Requirements:

Answer: The manuscript and figures have now been updated to meet PLOS ONE’s style requirements. 

2. Please note that all PLOS journals ask authors to adhere to our policies for sharing of data and materials: https://journals.plos.org/plosone/s/data-availability. According to PLOS ONE’s Data Availability policy, we require that the minimal dataset underlying results reported in the submission must be made immediately and freely available at the time of publication. As such, please remove any instances of 'unpublished data' or 'data not shown' in your manuscript and replace these with either the relevant data (in the form of additional figures, tables or descriptive text, as appropriate), a citation to where the data can be found, or remove altogether any statements supported by data not presented in the manuscript.

Answer: Any instances of ‘unpublished data’ have now been replaced with a reference to additional supplementary figures.

'E.E.: Swedish government project and salary funding for clinically oriented medical research (ALF-grants; F 2014/354) URL: https://www.skane.se/en/politics-and-organisation/research/services-and-support-for-researchers/grants-and-applications/

E.E.: Regional research and development grants (Southern healthcare region, Sweden; 170083): URL: https://www.skane.se/en/politics-and-organisation/research/services-and-support-for-researchers/grants-and-applications/

E.E.:The Crafoord Foundation (2017-0776) URL: https://en.crafoord.se/

M.J.H.:The Royal Physiographic Society in Lund (20141112) URL: https://www.fysiografen.se/en/

The funders had no role in study design, data collection and analysis, decision to publish, or preparation of the manuscript.'

We note that one or more of the authors are employed by a commercial company: NeuroVive Pharmaceutical AB

 Answer: This work was funded by Swedish government project and salary funding for clinically oriented medical research, Regional research and development grants (Southern healthcare region, Sweden;, The Crafoord Foundation and The Royal Physiographic Society in Lund). Additionally, this study was partially funded by NeuroVive Pharmaceutical AB (Lund, Sweden). NeuroVive Pharmaceutical provided support in the form of salaries for authors [S.P., J.K.E., I.C., E.E., M.J.H]. The specific roles of these authors are articulated in the ‘author contributions’ section. The funders did not have any additional role in the study design, data collection and analysis, decision to publish, or preparation of the manuscript.

We also note that you have a patent relating to material pertinent to this article.

In the amended statement of Competing Interests please declare this patent (with details including name and number), along with any other relevant declarations relating to employment, consultancy, patents, products in development or modified products etc.

Please confirm that this does not alter your adherence to all PLOS ONE policies on sharing data and materials, as detailed online in our guide for authors http://journals.plos.org/plosone/s/competing-interests by including the following statement: "This does not alter our adherence to PLOS ONE policies on sharing data and materials.” If there are restrictions on sharing of data and/or materials, please state these. Please note that we cannot proceed with consideration of your article until this information has been declared.

 Answer: S.P., J.K.E., I.C., F.S., E.E., and M.J.H have, or have had, salary from and/or equity interest in NeuroVive Pharmaceutical AB, a company active in the field of mitochondrial medicine. S.P., J.K.E., E.E., and M.J.H have filed patent applications for the use of succinate prodrugs for treatment of lactic acidosis or drug-induced side-effects due to complex I-related impairment of mitochondrial oxidative phosphorylation (WO/2015/155238) and protected carboxylic acid-based metabolites for treatment of mitochondrial disorders (WO/2017/060400, WO/2017/060418, WO/2017/060422). This does not alter our adherence to PLOS ONE policies on sharing data and materials.

Answer: The updated Funding Statement and Competing Interests Statement are now included in the cover letter and manuscript.

Answer: Any instances of ‘unpublished data’ have now been replaced with a reference to additional supplementary figures.

Reviewers' comments:

Reviewer's Responses to Questions

Comments to the Author

1. Is the manuscript technically sound, and do the data support the conclusions?

Reviewer #1: Yes

Reviewer #2: Partly

2. Has the statistical analysis been performed appropriately and rigorously?

Reviewer #1: Yes

Reviewer #2: Yes

3. Have the authors made all data underlying the findings in their manuscript fully available?

Reviewer #1: Yes

Reviewer #2: Yes

4. Is the manuscript presented in an intelligible fashion and written in standard English?

Reviewer #1: Yes

Reviewer #2: No

5. Review Comments to the Author

Review Comments to the Author

Reviewer #1: The manuscript entitled “Cell-permeable succinate prodrugs rescue mitochondrial respiration in cellular models of acute acetaminophen overdose” by Piel et al investigates the effects of acute acetaminophen exposure on mitochondrial function in primary human hepatocytes, HepG2 hepatocellular carcinoma cells, and platelets. The authors demonstrate that acetaminophen inhibits complex I, but not complex II function, and that treatment with a succinate prodrug (NV241) can rescue acetaminophen-induced mitochondrial dysfunction. Overall, the paper is very well written and clearly displays the negative respiratory outcomes associated with exposure to high concentrations of acetaminophen. The concept of utilizing succinate-based prodrugs to rescue pathologies associated with mitochondrial complex I dysfunction, particularly in an acute toxic exposure scenario, is an interesting area of research. There are a few minor points that should be addressed prior to publication.

1) One potential limitation in interpretation of the respiration graphs is that data are normalized to what I presume to be an initial seeding/plating density (i.e. cell number prior to treatment). Did the authors ever measure total protein or do some assessment of viability following acetaminophen and/or the mitochondrial inhibitor exposure? It would seem that the toxicity, particularly in the case of acetaminophen, could result in significant cell death from the starting number and thus influence the overall respiratory rates.

Answer: A valid question. As the reviewer correctly pointed out, direct measurements of cell-viability were not performed in this study. However, the increase in mitochondrial respiration in response to the cell-permeable succinate prodrug NV241, as shown in Figure 1-3, indicates intact cellular metabolism and the subsequent decrease in respiration due to antimycin A shows that this response is not unspecific chemical oxygen consumption but a biological response. The pharmacological prodrug strategy presented in this study requires intracellular metabolism to cleave off the two prodrug-moieties and release the active pharmaceutical ingredient, succinate [1]. Without intracellular metabolism, the prodrug is unable to release succinate and support mitochondrial respiration. Hence, an increase in response to the cell-permeable succinate prodrug is an indirect measure of cell viability. We demonstrate no difference between APAP-intoxicated cells and controls in response to the cell-permeable succinate prodrug NV241 with similar levels of mitochondrial respiration, indicating no difference in cell viability between the two groups. Therefore, direct measurements of cell viability were not further pursued in the present study. We will consider other approaches to assess cell viability for future studies investigating cell-permeable succinate prodrugs as treatment for APAP-induced liver damage.

2) In Figure 1, why were rotenone and NV241 added at the same time, whereas in Figures 2 and 3 NV241 was always added post-rotenone exposure? It would seem that the minimal effect of NV241 in the primary hepatocytes may be a result of the cotreatment blocking the rescue effect.

Answer: As rightly stated by the reviewer, primary human hepatocytes received rotenone and NV241 at the same time whereas HepG2 cells and human platelets received identical additions but in sequential order. The experiments with primary human hepatocytes were performed using the Seahorse XFe96 Analyzer (Agilent technologies, Massachusetts, USA), which allows only a total of four sequential additions per sample. The number of injections per sample is dictated by the equipment. The Seahorse XFe96 Analyzer was used for primary human hepatocytes because this instrument requires less amount of sample, which inherently is a problem when working with primary cells. In contrast, the high-resolution respirometer Oroboros-O2k (O2k, Oroboros Instruments, Innsbruck, Austria), used for the experiments with HepG2 cells and human platelets, where cell number is not a limiting factor, allows a nearly unlimited number of additions per sample. Therefore, it was possible to add rotenone and NV241 sequentially. While the mitochondrial respiration dependent on the electron transport system alone, measured following the addition of FCCP, was significantly different between APAP-intoxicated cells and controls (Fig. 1c), the respiration decreased down to the same level following the simultaneous addition of rotenone and NV241 (Fig. 1d). Hence, the simultaneous addition of rotenone with NV241 does not block the rescue effect of the cell-permeable succinate prodrug. Instead, the data show that the complex II-linked mitochondrial respiration supported by the cell-permeable succinate prodrug in the presence of rotenone (Fig. 1d) is unaffected by APAP. Therefore, for comparison of the treatment effect of NV241 between the different cell types not the response to the NV241 addition in itself should be evaluated. Instead the resulting/remaining CII-linked mitochondrial respiration in the presence of NV241 should be compared between APAP-intoxicated cells and controls. We have now clarified the corresponding results section of the manuscript.

3) It would seem that the reliance on oxidative phosphorylation might differ significantly between a primary hepatocyte, a liver cancer cell, and a circulating platelet. Did the authors take into account ECAR/glycolysis measurements in this or previous publications to account for changes in glycolytic flux? This could be covered in the discussion.

Answer: This is a good question, and the translatability between human platelets and tissue specific cell lines is something we continuously reevaluate. Platelets are an easily accessible and rich source of viable human mitochondria. They have previously been described to depend primarily on oxidative phosphorylation and have been regularly used for evaluation of mitochondrial function in disease and drug intoxication [2-5]. Also cancer cells, such as HepG2 cells, long believed to rely solely on glycolysis to meet their energy demand, have now been described to upregulate their mitochondrial metabolism under certain conditions and rely on mitochondrial function for several cancerogenic processes [6, 7]. Despite the difference in the absolute rate of oxidative phosphorylation the different cell types they showed similar relative sensitivity to APAP-induced mitochondrial toxicity: primary hepatocytes IC50 = 6.0 mM, HepG2 cells IC50 = 6.6 mM and human platelets IC50 = 7.4 mM (Fig. 4). The liver specific toxicity seen in patients with acute APAP intoxication is therefore likely due to the first-pass metabolism of APAP and not related to liver specific metabolism of the drug. This has now been addressed in the discussion.

4) In Figure 2B, it appears that two of the chosen wells are accounting for the significantly higher respiration in control HepG2 cells compared to the acetaminophen treated cells (even with the indicated significant). The authors should consider removing outliers or adding in additional wells to ensure the significant decrease in respiration that occurs following acetaminophen exposure in these cells.

Answer: We appreciate the attention to detail. Like the reviewer excellently spotted there were two replicates in Figure 2B which showed higher respiration than the remaining replicates of the experiment. We now have performed an outlier identification analysis. ROUT outlier analysis with a maximally desired false discovery rate of 1% detected no outliers in either the control or APAP group. Further data analysis showed that the difference among the six replicates is independent of APAP, as illustrated below. The two replicates which showed a higher routine respiration following exposure to vehicle or APAP also showed a higher routine respiration before start of exposure. Further, a high respiration of vehicle-treated cells matched a high respiration of the APAP-intoxicated cells of the same replicate, as now indicated by individual color coding of each replicate in the below figure. 

5) In Figure 5A, why does the FCCP titration taper off in the acetaminophen treated cells and then recover to control following rotenone treatment? Was this a result of negative readings/values? The authors could discuss in the results or discussion section.

Answer: We agree with the reviewer that the illustration in Figure 5A is misleading regarding the response to FCCP. The decrease in mitochondrial respiration following the last addition of FCCP in the APAP-treated cells is not a response to the protonophore FCCP but an artifact due to the injection in itself. The Oroboros O2k measures cellular respiration by calculating the negative derivative slope of the oxygen concentration in your sample. By injecting solutions to the sample, like in the described SUIT protocol, oxygen is simultaneously injected. This creates an artificial increase in oxygen concentration in the sample and hence, is interpreted by the software as a temporary decrease in respiration. This event is independent of the respiratory rate of the sample. Once the oxygen, that is injected with the solution, is equally distributed in the sample, the respiration signal is reflecting only the cellular respiration of the sample again. We agree with the reviewer that the selection where the mitochondrial respiration trace of the APAP-intoxicated sample was cut and adjusted to allow for comparison of the representative traces of both groups might be misleading. We have improved Figure 5A accordingly and included a clarification in the corresponding figure legend. 

Reviewer #2: While the manuscript submitted uses a variety of respirometry methods to accurately assess mitochondrial respiration (and therefore metabolism), as written, the manuscript makes conclusions that are "reaching" and not entirely supported by the data. Below are the major concerns that are outlined - these revision experiments would make the manuscript suitable for publication.

Major Concerns:

The authors present their succinate prodrug as an adjunctive therapy to NAC, yet do not do a single experiment comparing their prodrug to NAC, nor any combination therapy experiments. If the conclusion that this prodrug is adjunctive, they must show that in their experimental models.

Answer: A valid concern. The mechanism of action of N-acetylcysteine (NAC) and the novel, mitochondrial-targeted therapy of cell-permeable succinate prodrugs are very different. NAC protects liver cells from further oxidative damage by APAP by increasing glutathione levels and replenishing the cell’s antioxidant defense [8-10]. The cell-permeable succinate prodrug, in contrast, aims at rescuing already damaged cells by increasing mitochondrial respiration which is coupled to phosphorylation pathways and thereby potentially rescues the otherwise dying cells. Because both treatment strategies have a different mechanism of action and target different populations of liver cells in the situation of acute APAP intoxication we do think their treatment effects cannot be compared using the here described methodology: respirometry. We do agree with the reviewer that therefore our conclusions that the cell-permeable succinate prodrugs could be adjunctive therapies are too far reaching and have consequently removed these statements from the manuscript.

The authors did not assess the effect of their prodrug on any endpoints apart from mitochondrial respiration. While this reviewer appreciates the rationale (that this prodrug is specifically rescuing mitochondrial respiration), it would be interesting to see how this drug impacts downstream endpoints of cellular health that are affected by acetaminophen toxicity, such as ROS production, apoptosis/necrosis, cell proliferation, etc.

Pursuant to the previous comments, assessing downstream cell health endpoints after combination therapy would greatly strengthen the authors conclusions.

Answer: The suggestions by the reviewers are all very valid points, and we agree that the study could be further expanded this way. The compound NV241 is a first-of-its-kind succinate prodrug, previously reported as an investigative compound for primary mitochondrial disease [1]. It is a model drug of a first generation optimized for relatively short-term in vitro experiments, with very short half-life in plasma (minutes in humans, seconds in rodents). This makes the possible experimental designs somewhat limited. Oxidative damage and cellular death are cellular endpoints that have been reported to appear not earlier than 3-6 h following APAP ingestion and therefore would require drug candidates which demonstrate a longer stability [10]. Newer generations of compounds in the same pharmaceutical class, with better stability, are under investigation and will be applied to this project when possible. This is a clear limitation of this study and we have included a section on limitations of this study in the discussion part to reflect the lack of the above-mentioned models.

Minor Concerns:

The manuscript is generally well-written, however it would benefit from a thorough edit for syntax, use of commas, plurals, etc.

Answer: The reviewer is correct, there was room for improvement regarding the language quality. We have gone through the manuscript to further improve the language. 

1. Ehinger, J.K., et al., Cell-permeable succinate prodrugs bypass mitochondrial complex I deficiency. Nat Commun, 2016. 7: p. 12317.

2. Kramer, P.A., et al., Bioenergetics and the oxidative burst: protocols for the isolation and evaluation of human leukocytes and platelets. J Vis Exp, 2014(85).

3. Kramer, P.A., et al., A review of the mitochondrial and glycolytic metabolism in human platelets and leukocytes: implications for their use as bioenergetic biomarkers. Redox Biol, 2014. 2: p. 206-10.

4. Ravi, S., et al., Mitochondria in monocytes and macrophages-implications for translational and basic research. Int J Biochem Cell Biol, 2014. 53: p. 202-207.

5. Garcia-Souza, L.F. and M.F. Oliveira, Mitochondria: biological roles in platelet physiology and pathology. Int J Biochem Cell Biol, 2014. 50: p. 156-60.

6. Shiratori, R., et al., Glycolytic suppression dramatically changes the intracellular metabolic profile of multiple cancer cell lines in a mitochondrial metabolism-dependent manner. Scientific Reports, 2019. 9(1): p. 18699.

7. Porporato, P.E., et al., Mitochondrial metabolism and cancer. Cell Research, 2018. 28(3): p. 265-280.

8. Saito, C., C. Zwingmann, and H. Jaeschke, Novel mechanisms of protection against acetaminophen hepatotoxicity in mice by glutathione and N-acetylcysteine. Hepatology, 2010. 51(1): p. 246-54.

9. Lee, K.K., et al., Targeting mitochondria with methylene blue protects mice against acetaminophen-induced liver injury. Hepatology, 2015. 61(1): p. 326-36.

10. Hinson, J.A., D.W. Roberts, and L.P. James, Mechanisms of acetaminophen-induced liver necrosis. Handb Exp Pharmacol, 2010(196): p. 369-405.

---

## [Decision Letter · Decision Letter 1]

18 Mar 2020

Cell-permeable succinate prodrugs rescue mitochondrial respiration in cellular models of acute acetaminophen overdose

PONE-D-20-01070R1

Dear Dr. Piel:

We are pleased to inform you that your manuscript has been judged scientifically suitable for publication and will be formally accepted for publication once it complies with all outstanding technical requirements.

With kind regards,

Jianhua Zhang

Academic Editor

PLOS ONE

Additional Editor Comments (optional):

Reviewers' comments:

Reviewer's Responses to Questions

**Comments to the Author**

1. If the authors have adequately addressed your comments raised in a previous round of review and you feel that this manuscript is now acceptable for publication, you may indicate that here to bypass the “Comments to the Author” section, enter your conflict of interest statement in the “Confidential to Editor” section, and submit your "Accept" recommendation.

Reviewer #1: All comments have been addressed

Reviewer #2: All comments have been addressed

2. Is the manuscript technically sound, and do the data support the conclusions?

Reviewer #1: (No Response)

Reviewer #2: Yes

3. Has the statistical analysis been performed appropriately and rigorously? 

Reviewer #1: (No Response)

Reviewer #2: Yes

4. Have the authors made all data underlying the findings in their manuscript fully available?

Reviewer #1: (No Response)

Reviewer #2: Yes

5. Is the manuscript presented in an intelligible fashion and written in standard English?

Reviewer #1: (No Response)

Reviewer #2: Yes

6. Review Comments to the Author

Reviewer #1: (No Response)

Reviewer #2: (No Response)

7. PLOS authors have the option to publish the peer review history of their article (what does this mean?). If published, this will include your full peer review and any attached files.

Reviewer #1: No

Reviewer #2: No

---

## [Editor Report · Acceptance letter]

23 Mar 2020

PONE-D-20-01070R1 

Cell-permeable succinate prodrugs rescue mitochondrial respiration in cellular models of acute acetaminophen overdose 

Dear Dr. Piel:

I am pleased to inform you that your manuscript has been deemed suitable for publication in PLOS ONE. Congratulations! Your manuscript is now with our production department. 

With kind regards,

on behalf of

Dr Jianhua Zhang 

Academic Editor

PLOS ONE